# A New Type of Misaligned Journal Bearing with Flexible Structure

Woo-Ju Jeon [1] and Sung-Ho Hong [2],*

[1] LG Gasan Digital Center, LG Electronics, Seoul 07336, Republic of Korea; skymeche@hanmail.net
[2] Department of Mechanical System Engineering, Dongguk University-WISE Campus, Gyeongju-si 38066, Republic of Korea
* Correspondence: hongsh@dongguk.ac.kr; Tel.: +82-54-770-2211

**Abstract:** A flexible structure is applied to improve the lubrication performance of a misaligned journal bearing. The journal bearing is a representative sliding bearing, and there is damage due to metal-to-metal contact as a result of misalignment. Since misalignment is an unavoidable phenomenon, a journal bearing with a flexible structure was proposed as a way to improve it. The lubrication characteristics of the bearing were evaluated numerically under a steady-state condition. EHL (elastohydrodynamic lubrication) analysis considering elastic deformation was performed. The lubrication performance was compared in accordance with variation of the geometry of the flexible structure and evaluated based on the minimum film thickness. Moreover, the results of the journal bearing with a flexible structure were compared with those of the journal bearing without the flexible structure. The flexible structure was then applied in the form of a groove to the area supporting high load on the journal bearing; it was elastically deformed by the generated oil film pressure, which helps obscure a larger oil film. Through numerical analysis, it was found that the journal bearing with a flexible structure can improve the lubrication performance in the misaligned condition.

**Keywords:** elastohydrodynamic lubrication (EHL); flexible structure; minimum film thickness; misaligned journal bearing



## 1. Introduction

Hydrodynamic journal bearings are one of the representative types of sliding bearings. They are broadly used in high-speed rotating machinery and equipment [1]. The journal's axis is often misaligned in the bearing, and the film thickness for bearing clearance is distributed unevenly along the axial direction because of factors such as asymmetric loads or installation errors [2–4]. The axial asymmetric hydrodynamic pressure causes the location of the equivalent supporting point to deviate from the bearing axial mid-plane [5,6]. In recent years, misaligned journal bearings have been investigated with a great deal of interest. Jang and Khonsari [7] researched the cause of misalignments in journal bearings and studied their effects on the static and dynamic performances of the bearing. Pigott [8] showed that the load carrying capacity of the journal bearing decreases by 40% for a misalignment of about 0.0002 rad. Sun and Gui [9] showed that the oil film thickness is minimized at the bearing ends by the misalignment; they also reported that the minimum film thickness of a misaligned journal bearing was smaller than that of an aligned journal because of a decrease in the load carrying capacity [10]. When the film thickness between the bearing and the misaligned shaft decreases below a certain level, metal-to-metal contact can occur, which may cause a malfunction or breakdown of the system [11,12]. Shenoy and Pai [13] as well as Ram and Sharma [14,15] reported the combined effect of journal misalignment and wear on the bearing performance. In addition, Nikolakopoulos and Papadopoulos [12] constructed a numerical model to investigate the relationship between inclination angle and friction force, concluding that misalignment of the journal in the rotating machinery is one of the most important factors influencing bearing performance.

These studies generally assume that the temperature of the lubricant is constant. However, thermal effects need to be considered because increase in eccentricity, rotating speed, and friction loss can lead to a greater temperature rise of the lubricant [16]. He et al. [17] and Xu et al. [18] presented comprehensive analyses of the journal bearing performance considering the thermal effects; moreover, they found that the thermal effect had a major influence on the performance of the misaligned journal bearing when the eccentricity ratio was large. Sun et al. [9,10,19] reported the misalignment, thermal, and surface topography effects on bearing performance by using experimental and theoretical methods.

Many studies have been conducted on preventing contact problems due to misalignment of the journal bearing from various engineering perspectives. In terms of bearing material, Sharma et al. [20] applied graphite particles to ZA-27 alloy for the material of the journal bearing; as a result, the misaligned journal bearing (Figure 1a) could run without seizure. Kim et al. [21] presented reduction in the wear and friction coefficients using carbon-fiber phenolic composite as the material of the journal bearing. Moreover, lubricant development and profile applications to bearings were conducted to prevent contact problems in misaligned journal bearings. Das et al. [2] applied a micropolar fluid as the lubricant to the journal bearing to increase its load-carrying capacity and reduce the frictional force. Bouyer et al. [22] applied different profiles to the lubrication area at the bearing ends; they showed that the minimum film thickness increased by up to 60% in comparison with the journal bearing without the profile when the shaft was misaligned.

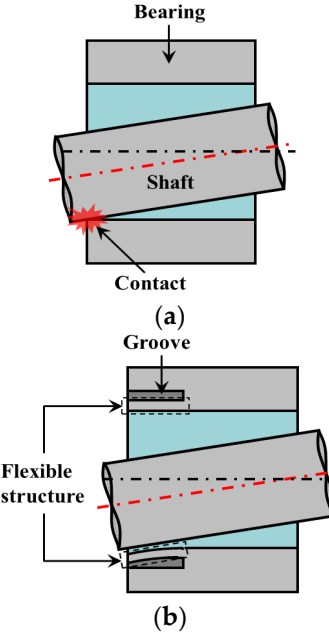

**Figure 1.** Application of the flexible structure to the misaligned journal bearing: (**a**) without flexible structure; (**b**) with flexible structure.

Furthermore, application of a flexible structure to the journal bearing is one method of preventing contact problems caused by a misalignment because the film thickness at the bearing end increases due to elastic deformation. The flexible structure is applied using a ring-shaped groove in the dotted region, as shown in Figure 1b. Elastic deformation occurs in the grooved zone due to the oil film pressure acting on the bearing surface, so it is possible to obtain a greater oil film thickness. Finally, the metal-to-metal contact caused by misalignment of the journal bearing can be prevented [23,24]. However, a flexible structure with inappropriate geometry may decrease the minimum film thickness [25].

Therefore, the design of an appropriate flexible structure is crucial for the misaligned journal bearing to effectively prevent contact problems and improve the lubrication performance. However, studies on the flexible structure of the misaligned journal bearing are not varied, and the results are only useful to specific applications, such as rotary and scroll

compressors. Furthermore, guidelines are not provided for an appropriate flexible structure for the journal bearing. Thus, it is difficult to design appropriate flexible structures to improve the lubrication characteristics of misaligned journal bearings. Numerical research on the various flexible structure geometries of misaligned journal bearings can be useful for designing appropriate flexible structures in various engineering fields.

In the present research, the influence of the flexible structure of the misaligned journal bearing is investigated under various misalignment conditions to provide guidelines regarding the flexible structure design. The dimensionless minimum film thickness is used to estimate the lubrication performances of misaligned journal bearings, which are compared with and without flexible structures.

## 2. Numerical Model and Method

The elastohydrodynamic lubrication (EHL) analysis is conducted to investigate the lubrication performance of the misaligned journal bearing, considering deformation in the flexible structure. In this study, the commercial software COMSOL version 6.0 is used for performing multi-physics analysis. The software uses a finite element method to discretize the non-linear governing equations to algebraic equations that can be solved iteratively. In the numerical analysis, the hydrodynamic bearing module is used for hydrodynamic lubrication analysis, the solid mechanics module is used for elastic deformation analysis of the bearing, and the solid-bearing coupling module is used for interworking between these two modules. Figure 2 shows the schematic of the journal bearing with radius $r$ and length $l$. The shaft is rigid and rotates around the $z$-axis with an angular velocity $\omega$. The $z$ and $x$ axes represent the axial and radial directions, respectively. The load w is applied in the direction of $\theta_W$ to the center of the shaft at the location where $z$ is equal to l/2. A ring-shaped groove is applied to the given section, as shown in Figure 2a,b. The flexible structure, which is shown in the dotted regions in Figure 2b,c, is applied to the bearing by the groove to obtain the elastic deformation. The geometry of the flexible structure is defined by the length (lf), inner end thickness (d), and outer end thickness (a). The sectional shape in the axial direction of the flexible structure depends on $\gamma$ (= $d/a$) and is rectangle- (for $\gamma = 1$) or taper-shaped (for $\gamma > 1$). The bearing surface where the groove is not applied is rigid because the thickness is large; therefore, elastic deformation does not occur. The dimensionless parameters for the geometries of the journal bearing and the flexible structure are defined in Equation (1).

$$A = \frac{a}{r}, \; L = \frac{l}{r}, \; L_f = \frac{l_f}{l}, \; \beta = \frac{c}{r}, \; \gamma = \frac{d}{a} \tag{1}$$

where, $A$, $L$, $L_f$, $\beta$, and $\gamma$ express dimensionless thickness of the flexible structure, ratio of length to radius of the bearing, dimensionless length of the flexible structure, ratio of clearance to bearing radius, and ratio of thickness at both ends of the flexible structure, respectively.

The Reynolds equation shown in Equation (2) is used to solve the oil film pressure p.

$$\frac{1}{r^2} \frac{\partial}{\partial \theta} \left( h^3 \frac{\partial p}{\partial \theta} \right) + \frac{\partial}{\partial z} \left( h^3 \frac{\partial p}{\partial z} \right) = 6 \eta \omega \frac{\partial h}{\partial \theta} \tag{2}$$

where, $r$, $z$, and $\theta$ express cylindrical coordinates. Moreover, $h$, $p$, $\eta$, and $\omega$ represent oil film thickness, oil film pressure, absolute viscosity of the lubricant, and angular velocity, respectively.

The dimensionless form of Equation (2) can be written as

$$\frac{\partial}{\partial \theta} \left( H^3 \frac{\partial P}{\partial \theta} \right) + \frac{\partial}{\partial Z} \left( H^3 \frac{\partial P}{\partial Z} \right) = \frac{\partial H}{\partial \theta} \tag{3}$$

where $H$, $P$, and $Z$ represent dimensionless oil film thickness, dimensionless oil film pressure, and dimensionless rectangular coordinate system in the $z$ direction, respectively.

$$H = \frac{h}{c}, \ P = \frac{c^2(p - p_a)}{6r^2\eta\omega}, \ X = \frac{x}{r}, \ Y = \frac{y}{r}, \ Z = \frac{z}{r} \tag{4}$$

The parameter, $p_a$ is the atmospheric pressure.

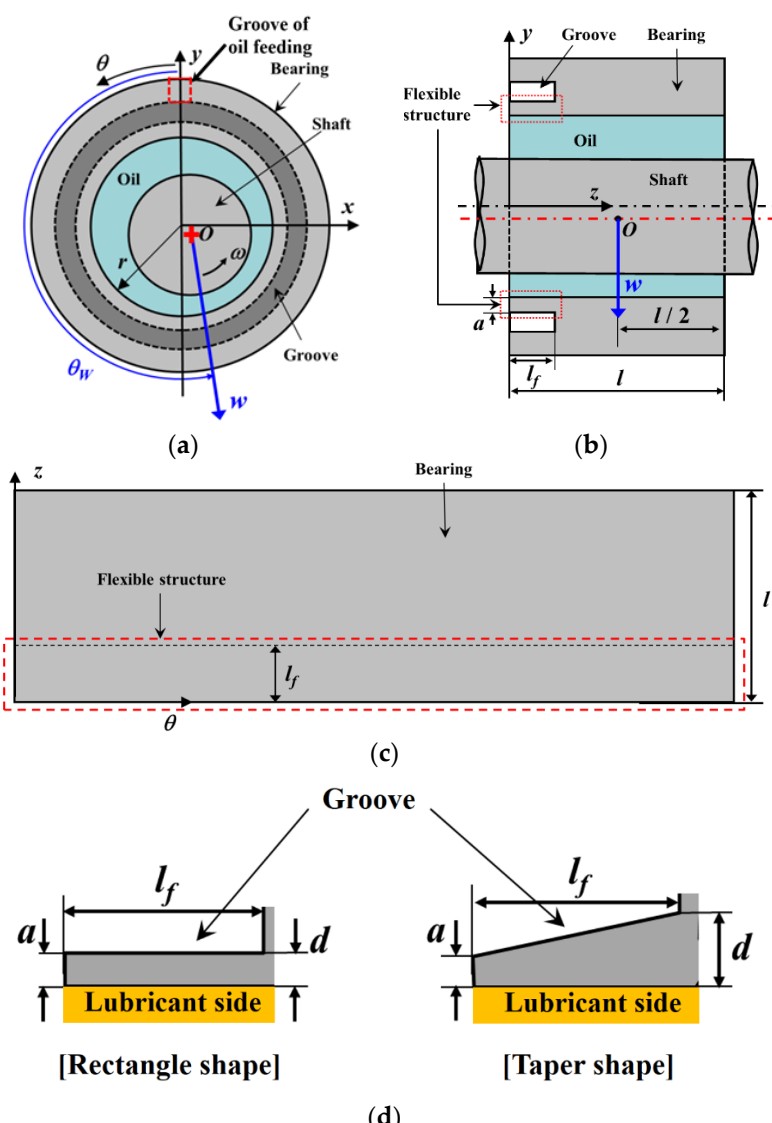

**Figure 2.** Schematics of the journal bearing system: (**a**) $x$–$y$ plane; (**b**) $y$–$z$ plane; (**c**) $\theta$–$z$ plane; (**d**) shapes of the flexible structure.

The oil film thickness $h$ between the bearing and shaft should be obtained to calculate the oil film pressure in Equation (2). The shaft rotates the bearing with eccentric and tilted motions, as shown in Figure 3; $O$ is the center of the shaft at the location where $z$ is equal to $l/2$. $O_1$, $O_2$ are the centers of the shaft at both ends of the bearing, respectively. Two circles in Figure 3a are cross sections of the shaft projected onto the $x$–$y$ plane, and $O_1$, $O_2$ are their centers. The eccentric amount, $e$, is a distance on the $x$–$y$ plane between $O$ and the center of the bearing. The attitude angle $\psi$ is an angle on the $x$–$y$ plane between the load direction and a straight line passing through $O$ and the bearing center. The tilting direction of the shaft is equal to $\theta_W$, and the tilting amount $e'$ is equal to half the distance between $O_1$ and $O_2$ on the $x$–$y$ plane. When the tilting amount is equal to zero, the shaft rotates in the

bearing under the aligned condition. The oil film thickness between the shaft and bearing is calculated using Equation (5).

$$h = c + e \cdot cos\{\theta - (\theta_W - \pi + \psi)\} + e\prime\left(1 - \frac{2}{l}z\right)cos\{\theta - (\theta_W - \pi)\} + h_e \tag{5}$$

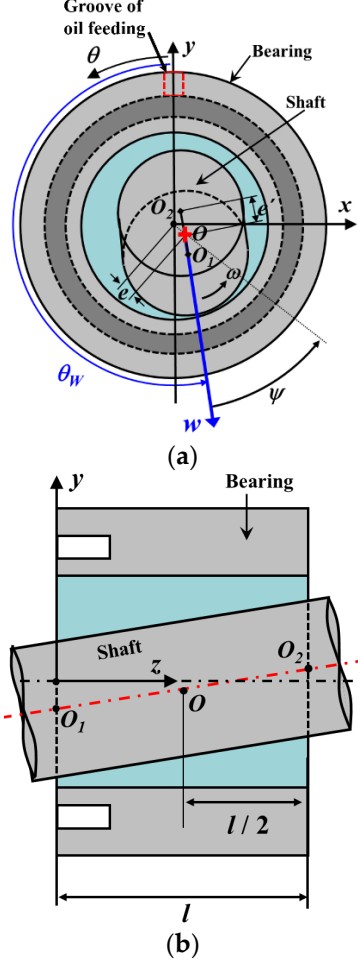

**Figure 3.** The shaft in the bearing with eccentric and tilted motion: (**a**) *x–y* plane; (**b**) *y–z* plane.

The dimensionless form of Equation (5) can be written as

$$H = 1 + \varepsilon \cdot cos\{\theta - (\theta_W - \pi + \psi)\} + \varepsilon\prime\left(1 - 2\frac{Z}{L}\right)cos\{\theta - (\theta_W - \pi)\} + H_e \tag{6}$$

where

$$H_e = \frac{h_e}{c}, \ \varepsilon = \frac{e}{c}, \ \varepsilon\prime = \frac{e\prime}{c} \tag{7}$$

The parameters $\varepsilon$ and $\varepsilon'$ in Equation (7) are the eccentricity and tilting ratios of the shaft, respectively. The oil film thickness changes with elastic deformation of the flexible structure, which should also be obtained to calculate Equation (5). The elastic deformation of the bearing is calculated using the solid mechanics module, and the numerical results are interworked with the hydrodynamic bearing module for hydrodynamic lubrication analysis. Figure 4 shows the hexahedral mesh of the finite element model. The hexahedral mesh is used for numerical calculations owing to its higher resolution and faster calculation in comparison to the tetrahedral mesh [26]. A mesh validation was performed, and the difference in the results' values was less than 2%. In the analysis, 52 elements are applied in the circumferential direction, and 53 elements are applied in the axial direction, except in the groove region. In the groove region, 108 elements are applied in the circumferential

direction. The total number of elements is thus 50,220. The value of the convergence criteria is given as $10^{-3}$ for continuity, velocity, and deformation.

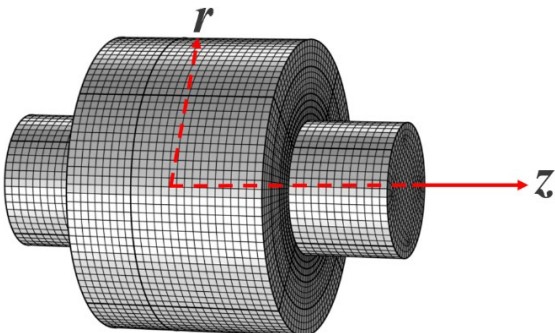

**Figure 4.** 3-dimensional finite element analysis model.

The boundary conditions to solve Equations (2) and (8) are defined as follows:

■　Boundary condition for the oil film fracture zone:

$$\frac{\partial p}{\partial n} = 0, \; p = p_a$$

■　Pressure in the cavitation region:

$$p = p_a \; \text{where} \; p < p_a \tag{8}$$

■　Pressure at bearing ends and oil feeding groove:

$$p(0, z) = p_b, \; p(\theta, 0) = p_b, \; p(\theta, l) = p_b$$

■　Displacement at the inner end of the flexible structure:

$$u\left(x, y, l_f\right) = 0$$

The direction of $n$, which is shown in Equation (8), is perpendicular to the oil film fracture boundary line. Periodic conditions of pressure and displacement are also used for the ring shape of the bearing. Equation (8) in the dimensionless form can be written as Equation (9).

■　Boundary condition for the oil film fracture zone:

$$\frac{\partial P}{\partial n} = 0, \; P = 0$$

■　Pressure in the cavitation region

$$P = 0 \; \text{where} \; P < 0 \tag{9}$$

■　Pressure at the bearing ends and oil feeding groove:

$$P(0, Z) = P_b, \; P(\theta, 0) = P_b, \; P(\theta, L) = P_b$$

■　Displacement at the inner end of the flexible structure:

$$U\left(X, Y, L_f\right) = 0$$

The oil film force $f_o$ is generated by the oil film pressure and is used to calculate the motion of the shaft by comparing with the load with the load. Figure 5 shows the oil film

force acting on the shaft. The parameters $f_{ox}, f_{oy}$ are components of the oil film force in the $x, y$ directions and are calculated using Equations (10) and (11), respectively. The oil film force is calculated with Equation (12).

$$f_{ox} = \int_0^l \int_0^{2\pi} p\, r\, sin\theta d\theta dz \tag{10}$$

$$f_{oy} = -\int_0^l \int_0^{2\pi} p\, r\, cos\theta d\theta dz \tag{11}$$

$$f_o = \sqrt{f_{ox}^2 + f_{oy}^2} \tag{12}$$

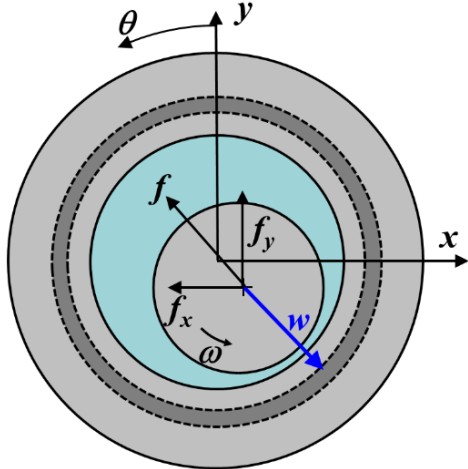

**Figure 5.** Oil film force working on the shaft.

Equations (10)–(12) are written in dimensionless form as follows:

$$F_{OX} = \int_0^L \int_0^{2\pi} P\, sin\theta d\theta dZ \tag{13}$$

$$F_{OY} = -\int_0^L \int_0^{2\pi} P\, cos\theta d\theta dZ \tag{14}$$

$$F_O = \sqrt{F_{OX}^2 + F_{OY}^2} \tag{15}$$

$$F_{OX} = \frac{c^2 f_{ox}}{6r^4 \eta \omega}, \ F_{OY} = \frac{c^2 f_{oy}}{6r^4 \eta \omega}, \ F_O = \frac{c^2 f_o}{6r^4 \eta \omega} \tag{16}$$

## 3. Numerical Results and Discussion

EHL analysis [27–32] was conducted to estimate the lubrication characteristics of the misaligned journal bearing with flexible structure under the specifications shown in Table 1. Figures 6 and 7 show the dimensionless film thickness and oil film pressure characteristics of the no-groove and grooved journal bearings, respectively. The grooved journal bearing includes the flexible structure, and the dimensionless length of the structure is equal to $L_f$. This means that the flexible structure does not apply when the dimensionless length is equal to zero because the bearing is not grooved. Figures 6a and 7a show the dimensionless oil film thickness distributions of the no-groove and grooved journal bearings, respectively. Elastic deformation occurs and the dimensionless oil film thickness increases due to the flexible structure in the region between 0 and 1 in the $Z$ direction, as shown in Figure 7a, when compared with the results of the no-groove journal bearing in Figure 6a. The dimensionless oil film pressure distributions of the no-groove and grooved journal bearings

are shown in Figures 6b and 7b, respectively. Moreover, the dark-blue areas represent the cavitation regions in these figures. The dimensionless oil film pressure in the flexible region of the grooved journal bearing decreases due to the elastic deformation owing to the larger film thickness, and the location of the peak pressure in the $Z$ direction moves to the inside of the bearing in comparison to the no-groove one. Figures 6c and 7c are the circumferential distributions of the dimensionless film thickness and pressure at three lined locations in Figure 6a,b and Figure 7a,b, respectively. When the flexible structure is applied to the misaligned journal bearing, the film thickness increases and oil film pressure decreases in the flexible region owing to elastic deformation, as shown in Figure 7c, in comparison to Figure 6c. In addition, the eccentricity ratio $\varepsilon'$ increases when the flexible structure is applied under a constant load because the load capacity decreases with increase in oil film thickness.

**Table 1.** Specification of the analysis model (application of flexible structure).

| Parameter | Value | Parameter | Value |
|---|---|---|---|
| $A$ | 0.4 | $W$ | 4.4 |
| $E^*$ | $2.2 \times 10^4$ | $\beta$ | $10^{-3}$ |
| $L$ | 3.0 | $\gamma$ | 1.0 |
| $L_f$ | 0, 1/3 | $\varepsilon'$ | 0.2 |
| $P_b$ | 0.3 | | |

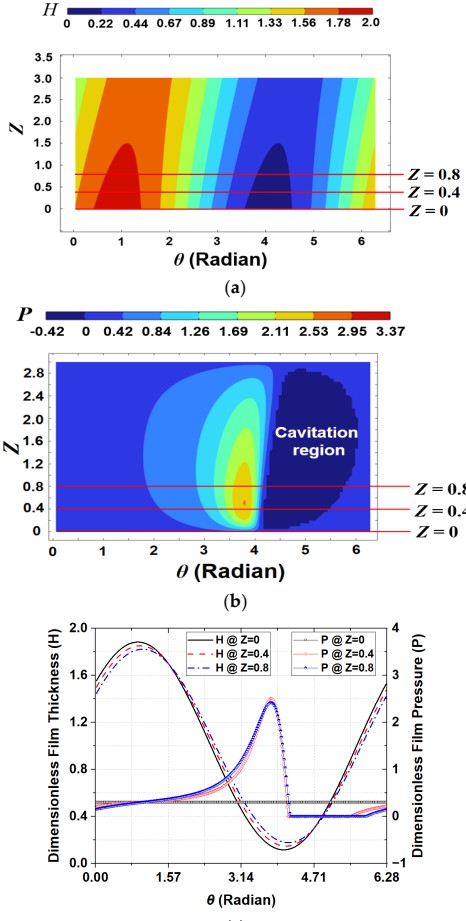

**Figure 6.** Dimensionless results for journal bearing without flexible structure ($L$ = 3.0, $W$ = 4.4, $\gamma$ =1.0, $\varepsilon'$ = 0.2): (**a**) film thickness; (**b**) pressure distribution; (**c**) circumferential distribution of film thickness and pressure.

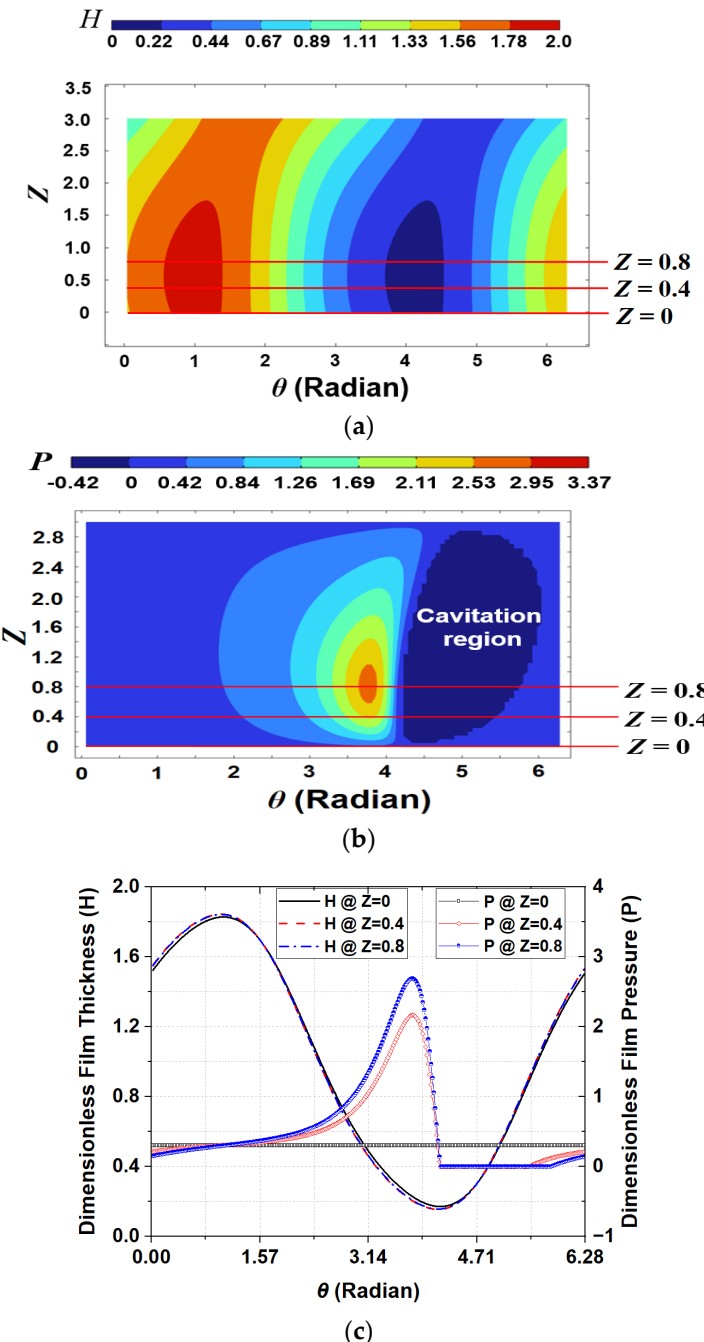

**Figure 7.** Dimensionless results for journal bearing with flexible structure ($L = 3.0$, $A = 0.4$, $W = 4.4$, $\gamma = 1.0$, $\varepsilon' = 0.2$): (**a**) film thickness; (**b**) pressure distribution; (**c**) circumferential distribution of film thickness and pressure.

The main purpose of the flexible structure is to prevent contact at the misaligned journal bearing end. Therefore, the minimum film thicknesses required for various misalignments and flexible structure geometries need to be compared. First, the minimum film thicknesses for different tilting ratios are compared with those of the no-groove journal bearing under a constant flexible structure geometry, and these specifications are shown in Table 2. Figure 8 shows the dimensionless minimum film thickness ($H_{min}$) with variation of the tilting ratio $\varepsilon'$. The dimensionless minimum film thickness decreases when the tilting ratio increases under a constant load. The no-groove journal bearing has a dimensionless minimum film thickness of about 0.03 when the tilting ratio is equal to 0.4. This value is 3% of the clearance and may cause metal-to-metal contact at the bearing end because

the minimum film thickness is significantly small. The journal bearing with flexible structure obtains a larger dimensionless minimum film thickness owing to the increase in the film thickness due to elastic deformation as compared to the no-groove one. Moreover, the amount of increase in the dimensionless minimum film thickness due to the flexible structure is proportional to the tilting ratio. The dimensionless minimum film thickness of the bearing with flexible structure is about 2.3 times that of the no-groove one for the case where the tilting ratio is equal to 0.4.

**Table 2.** Specification of the analysis model (variation of $\varepsilon'$).

| Parameter | Value | Parameter | Value |
|---|---|---|---|
| $A$ | 0.4 | $W$ | 4.4 |
| $E^*$ | $2.2 \times 10^4$ | $\beta$ | $10^{-3}$ |
| $L$ | 3.0 | $\gamma$ | 1.0 |
| $L_f$ | 0, 1/3 | $\varepsilon'$ | 0.1, 0.2, 0.3, 0.4 |
| $P_b$ | 0.3 | $\theta_w$ | $\pi$ |
| $\nu$ | 0.3 | | |

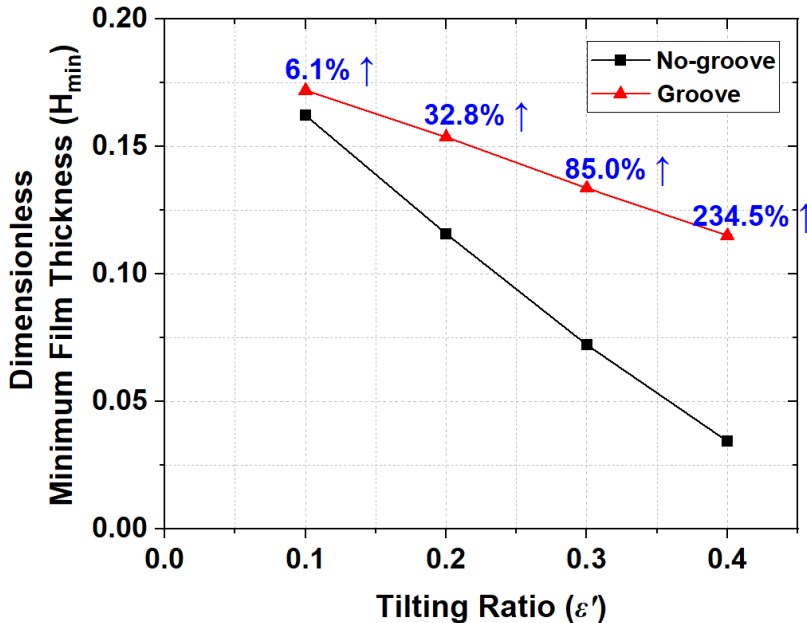

**Figure 8.** Dimensionless minimum film thickness with shaft tilting ratio ($L$ = 3.0, $A$ = 0.4, $L_f$ = 1/3, $W$ = 4.4, $\gamma$ = 1.0).

The dimensionless minimum film thickness of the misaligned journal bearing increases with application of the flexible structure. In addition, the minimum film thickness increases more when the shaft is more misaligned. That is, the flexible structure of the misaligned journal bearing can effectively prevent metal-to-metal contact at the bearing end.

The influences of the flexible structure geometry change can also be estimated to suggest a guideline for appropriate flexible structure design. Therefore, analysis of variations of the flexible structure geometry is carried out. The geometry of the flexible structure is varied by changing the dimensionless thickness ($A$), length ($L_f$), and thickness ratio ($\gamma$).

The elastic deformation is a function of the thickness of the flexible structure and influences the lubrication performance of the misaligned journal bearing. Figure 9 shows the dimensionless minimum film thickness with variation of the dimensionless thickness of the flexible structure, as shown in Table 3. The analysis is also performed for different tilting ratios as shown in Figure 9a–c.

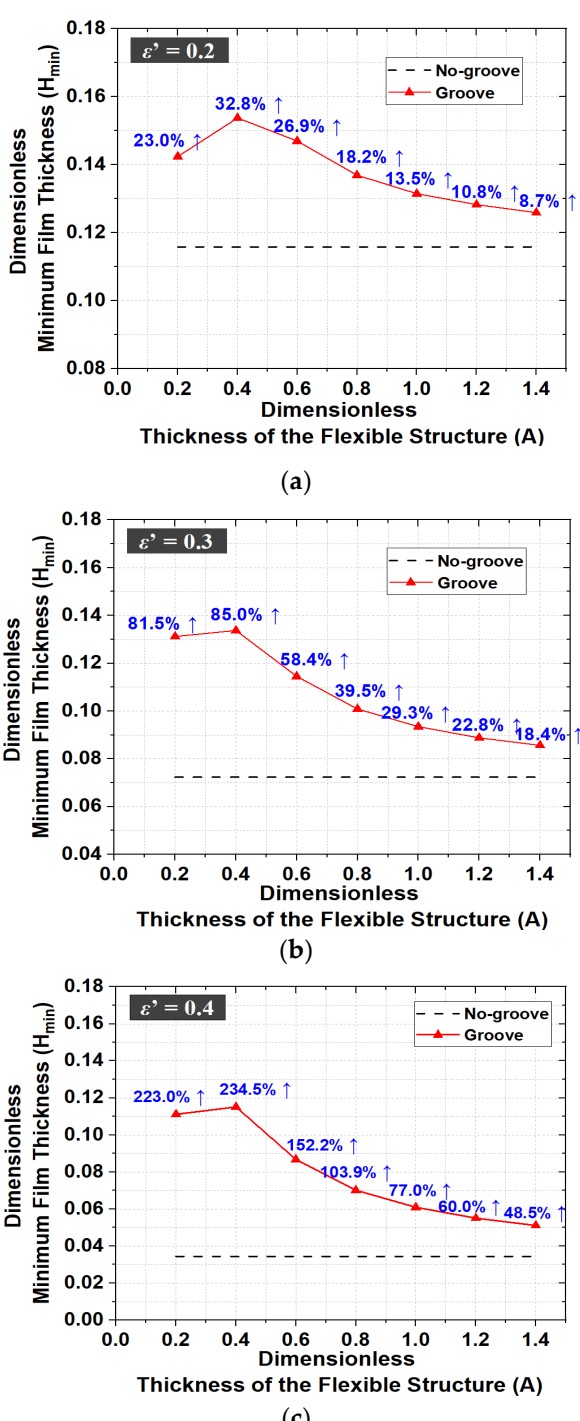

**Figure 9.** Dimensionless minimum film thickness with $A$ ($L = 3.0$, $L_f = 1/3$, $W = 4.4$, $\gamma = 1.0$). (**a**) $\varepsilon' = 0.2$; (**b**) $\varepsilon' = 0.3$; (**c**) $\varepsilon' = 0.4$.

**Table 3.** Specification of the analysis model (variation of $A$).

| Parameter | Value | Parameter | Value |
|:---:|:---:|:---:|:---:|
| $A$ | 0.2~1.2 | $W$ | 4.4 |
| $E^*$ | $2.2 \times 10^4$ | $\beta$ | $10^{-3}$ |
| $L$ | 3.0 | $\gamma$ | 1.0 |
| $L_f$ | 0, 1/3 | $\varepsilon'$ | 0.2, 0.3, 0.4 |
| $P_b$ | 0.3 | $\theta_w$ | $\pi$ |
| $\nu$ | 0.3 | | |

The values of $Z$ in Figure 9 indicate the locations along the $Z$ direction of the dimensionless minimum film thickness. The location is the bearing end when $Z$ is equal to zero, and the inner end of the flexible structure between the flexible and rigid regions of the bearing surface is represented when $Z$ is not zero. When the tilting ratio is equal to 0.2 and the dimensionless thickness of the flexible structure ($A$) is equal to 0.4, the dimensionless minimum film thickness is maximized. Compared to the case without the flexible structure, the difference in the dimensionless minimum film thickness is about 32%. The dimensionless minimum film thickness is also maximized when the dimensionless thickness of the flexible structure is equal to 0.4 for other tilting ratios, as shown in Figure 9b,c. When the tilting ratio increases, the ratio of increase in minimum film thickness with the flexible structure is greater than that for the case without the flexible structure. It can be seen that as the shaft of the bearing tilts more, the lubrication performance by the flexible structure is improved.

Figure 10 shows the circumferential distributions of the dimensionless film thickness along three different lines, such as the results in Figures 6 and 7 when the tilting ratio is equal to 0.3. In the dotted areas of these figures, the oil film pressure generation and deformation occur. The dimensionless minimum film thickness is maximized when the dimensionless thickness of the flexible structure is equal to 0.4. Moreover, the distributions of the dimensionless film thickness in the dotted areas are almost constant along the axial direction, and the dimensionless film thicknesses are relatively large when the dimensionless thickness of the flexible structure is equal to 0.4.

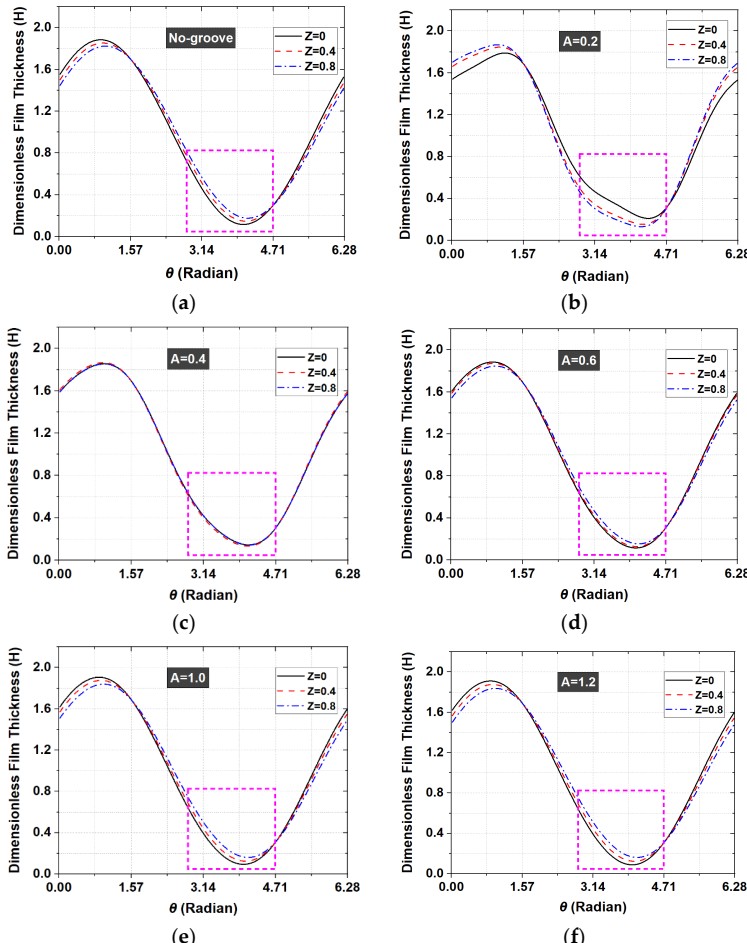

**Figure 10.** Circumferential distributions of the dimensionless film thickness for different $A$ ($L = 3.0$, $L_f = 1/3$, $W = 4.4$, $\gamma = 1.0$, $\varepsilon' = 0.3$). (**a**) No-groove; (**b**) $A = 0.2$; (**c**) $A = 0.4$; (**d**) $A = 0.6$; (**e**) $A = 1.0$; (**f**) $A = 1.2$.

The appropriate dimensionless thickness of the flexible structure can be used to maximize the dimensionless minimum film thickness. Moreover, to obtain a larger film thickness, it is necessary to find an appropriate thickness of the flexible structure that can be used as a reference because the location of the minimum film thickness in the *Z* direction changes when the thickness of the flexible structure varies.

The influence on lubrication performance with variation in the dimensionless length of the flexible structure ($L_f$) is also investigated. Figure 11 shows the dimensionless minimum film thickness with variation in the dimensionless length of the flexible structure and tilting ratios. The geometries and their specifications are shown in Table 4.

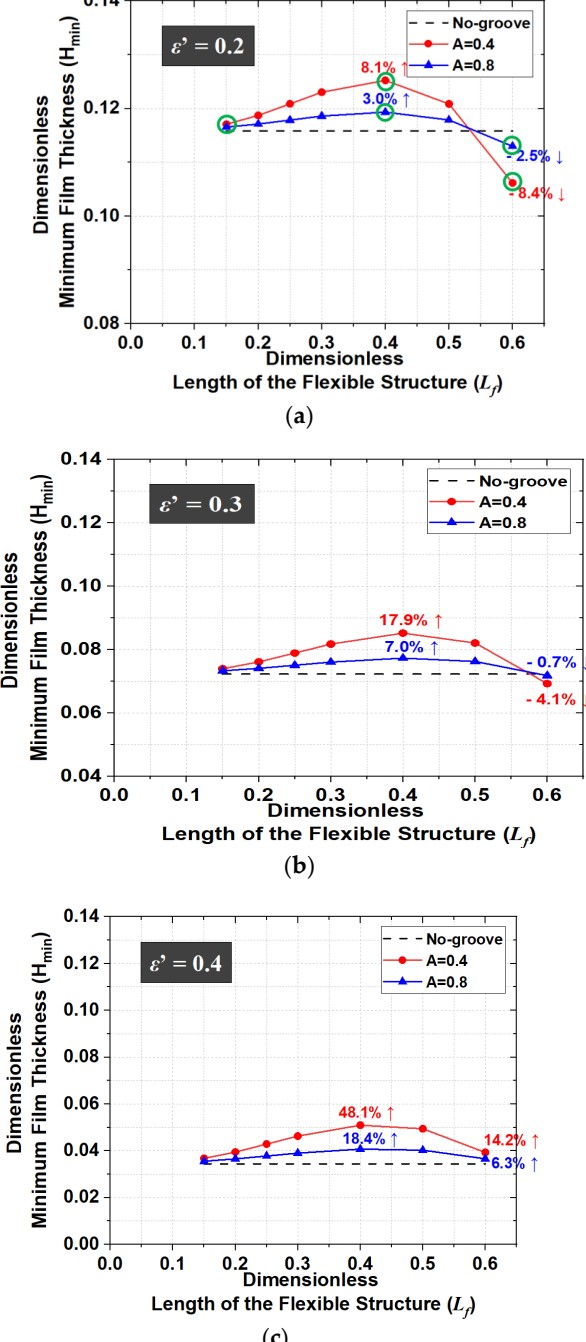

**Figure 11.** Dimensionless minimum film thickness with $L_f$ ($L$ = 3.0, $W$ = 4.4, $\gamma$ =1.0). (**a**) $\varepsilon'$ = 0.2; (**b**) $\varepsilon'$ = 0.3; (**c**) $\varepsilon'$ = 0.4.

**Table 4.** Specification of the analysis model (variation of $L_f$).

| Parameter | Value | Parameter | Value |
|---|---|---|---|
| $A$ | 0.4, 0.8 | $W$ | 4.4 |
| $E^*$ | $2.2 \times 10^4$ | $\beta$ | $10^{-3}$ |
| $L$ | 3.0 | $\gamma$ | 1.0 |
| $L_f$ | $0 \sim 1/3$ | $\varepsilon'$ | 0.2, 0.3, 0.4 |
| $P_b$ | 0.3 | $\theta_\mathrm{w}$ | $\pi$ |
| $\nu$ | 0.3 | | |

As the dimensionless length of the flexible structure increases, the dimensionless minimum film thickness increases up to a certain point and then decreases, as shown in Figure 11a. When the dimensionless length of the flexible structure is 0.6 for $\varepsilon' = 0.2$, the dimensionless minimum film thickness with the flexible structure is smaller than that without. Moreover, as the dimensionless length of the flexible structure increases, the dimensionless minimum film thickness for $A = 0.4$ is generally larger than that for $A = 0.8$. When the tilting ratio increases, the dimensionless length of the flexible structure corresponding to the region where the dimensionless minimum film thickness in the case with the flexible structure is larger than that without is longer. Figures 12 and 13 show the dimensionless pressure distributions and deformations for six cases, respectively. These six cases are marked with green circles in Figure 11a, showing the results for a tilting ratio of 0.2. In Figures 12a–c and 13a–c, as the dimensionless length of the flexible structure increases, the oil pressure increases, and the amount of elastic displacement accordingly increases. Such a tendency is also shown in Figures 12d–f and 13d–f. This means that as the length of the flexible structure increases beyond a certain level, the amount of elastic deformation also increases, but the minimum film thickness decreases. In other words, if the length of the flexible structure is inappropriate or excessive, the lubrication performance is worse than it is without the flexible structure. Therefore, to improve the lubrication performance in the misaligned journal bearing, it is necessary to find and apply the appropriate length of the flexible structure.

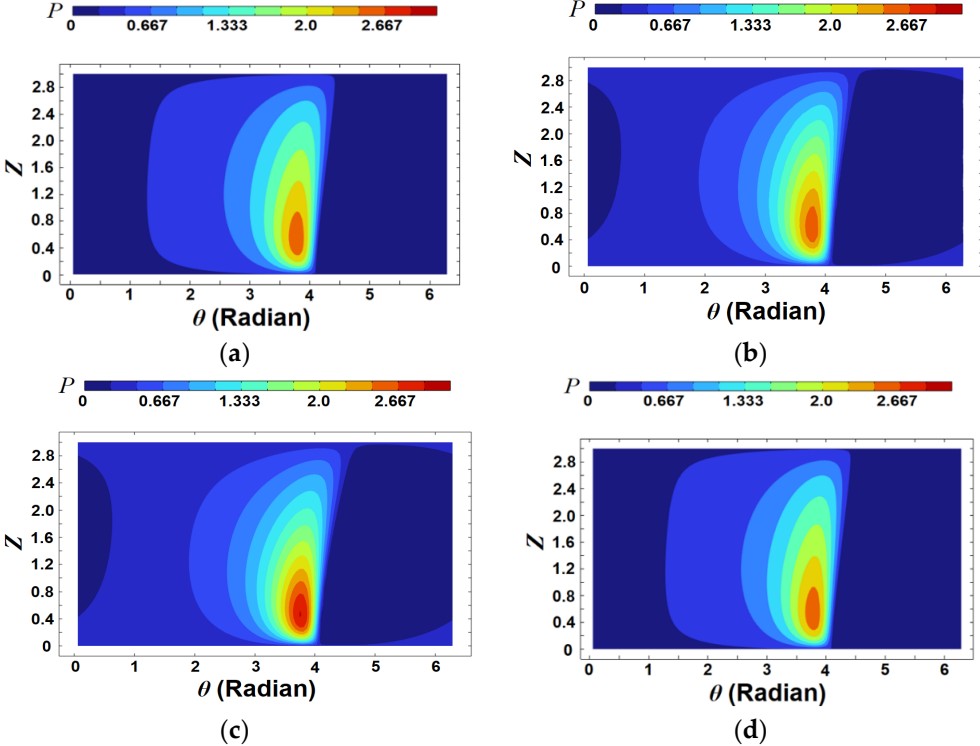

**Figure 12.** *Cont.*

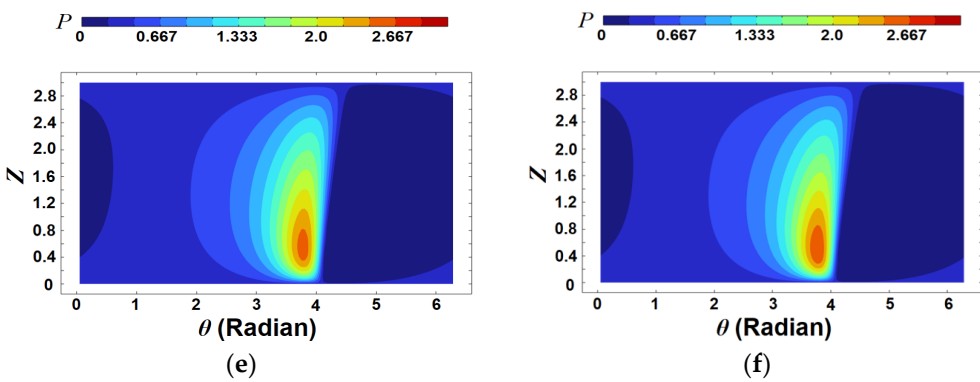

**Figure 12.** Distributions of the dimensionless oil pressure for $\varepsilon' = 0.2$. (**a**) $A = 0.4$, $L_f = 0.15$; (**b**) $A = 0.4$, $L_f = 0.4$; (**c**) $A = 0.4$, $L_f = 0.6$; (**d**) $A = 0.8$, $L_f = 0.15$; (**e**) $A = 0.8$, $L_f = 0.4$; (**f**) $A = 0.8$, $L_f = 0.6$.

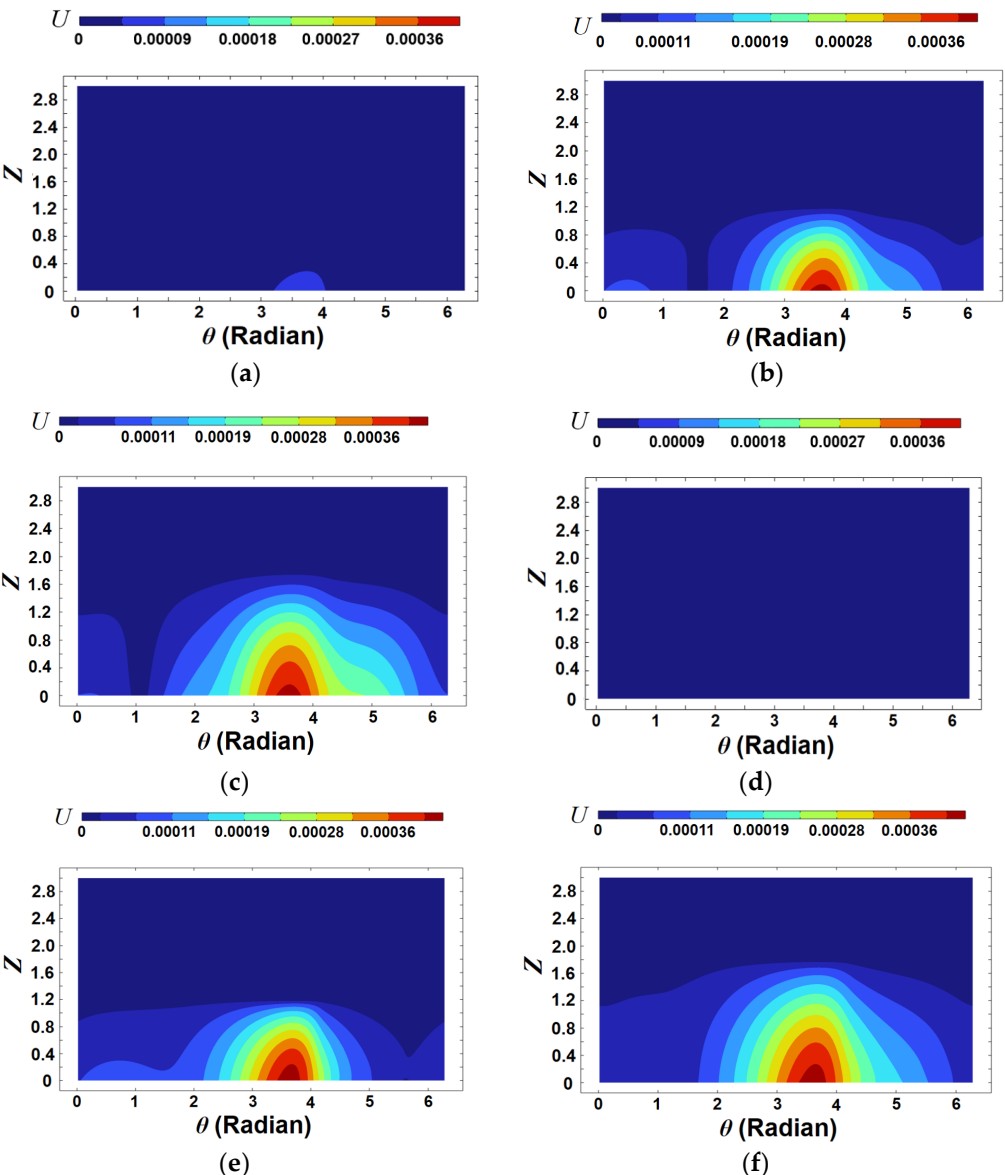

**Figure 13.** Distributions of the dimensionless deformation for $\varepsilon' = 0.2$. (**a**) $A = 0.4$, $L_f = 0.15$; (**b**) $A = 0.4$, $L_f = 0.4$; (**c**) $A = 0.4$, $L_f = 0.6$; (**d**) $A = 0.8$, $L_f = 0.15$; (**e**) $A = 0.8$, $L_f = 0.4$; (**f**) $A = 0.8$, $L_f = 0.6$.

The dimensionless minimum film thickness is maximized when the film thickness distribution in the area that is flexible and where the oil film pressure is generated is almost

constant along the axial direction. Then, the elastic deformation amount at the outer region of the flexible structure should be larger than that of the inner region for the misaligned journal bearing. Therefore, the analysis results for various cross-sectional shapes of the flexible structure are also estimated for the specifications shown in Table 5. The minimum film thicknesses for various thickness ratios of the flexible structure ends ($\gamma$) are compared with those of the no-groove journal bearing, and the results are shown in Figure 14.

**Table 5.** Specification of the analysis model (variation of $\gamma$).

| Parameter | Value | Parameter | Value |
|---|---|---|---|
| $A$ | 0.2, 0.4 | $W$ | 4.4 |
| $E^*$ | $2.2 \times 10^4$ | $\beta$ | $10^{-3}$ |
| $L$ | 3.0 | $\gamma$ | 1.0~5.0 |
| $L_f$ | 0~1/3 | $\varepsilon'$ | 0.2, 0.3, 0.4 |
| $P_b$ | 0.3 | $\theta_w$ | $\pi$ |
| $\nu$ | 0.3 | | |

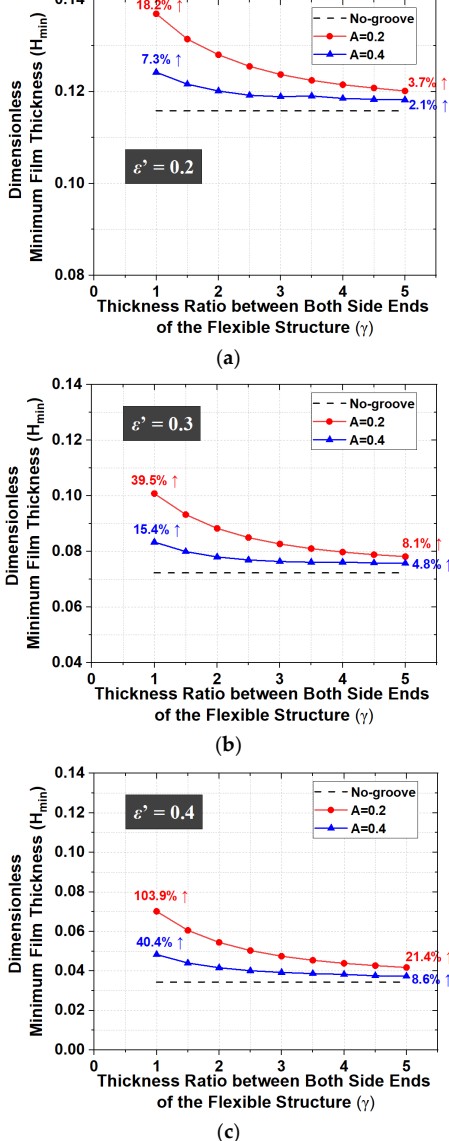

**Figure 14.** Dimensionless minimum film thickness with $\gamma$ ($L$ = 3.0, $W$ = 4.4, $\gamma$ =1.0). (**a**) $\varepsilon'$ = 0.2; (**b**) $\varepsilon'$ = 0.3; (**c**) $\varepsilon'$ = 0.4.

When the dimensionless thickness of the flexible structure is equal to 0.2, the dimensionless minimum film thicknesses are larger than those for $A = 0.4$. The reason for this is that the smaller dimensionless thickness of the flexible structure enables a larger oil film thickness owing to deformation by the oil film pressure. When $\gamma$ changes, the dimensionless minimum film thickness in the case of $\varepsilon' = 0.2$ is larger than those for $\varepsilon' = 0.3$ and $\varepsilon' = 0.4$. However, when the flexible structure is applied, the dimensionless minimum film thickness increases by 18.2% ($\varepsilon' = 0.2$), 39.5% ($\varepsilon' = 0.3$), and 103.9% ($\varepsilon' = 0.4$) compared to the values of the no-groove journal bearing. That is, the increase in the dimensionless minimum film thicknesses with the flexible structure relative to those without the flexible structure is relatively large when the tilting ratio is 0.4. In addition, the rectangular flexible structure is more effective for increasing the minimum film thickness than the taper-shaped one for all tilting ratios. The reason for this is that it is not easy to increase the oil film thickness owing to elastic deformation of the inside of the flexible structure when $\gamma$ increases. Therefore, to improve the lubrication characteristics in the misaligned journal bearing, the shape of the flexible structure should be designed such that the elastic deformation of the outer region of the flexible structure is larger than that of the inner region.

## 4. Conclusions

This study suggests the application of a flexible structure to misaligned journal bearings to improve their lubrication performances. EHL analysis was conducted to consider the deformation of the groove region using commercial software that can perform multiphysics analysis. The lubrication performance was estimated by comparing the minimum film thickness of the journal bearing with a flexible structure for variation of the dimensionless thickness, length, end thickness ratios of the flexible structure at both ends, and tilting ratio. In addition, these results were compared with those of the journal bearing without the flexible structure. The numerical analysis results indicate the following:

1. The flexible structure of the misaligned journal bearing increases the minimum film thickness effectively because the film thicknesses in the flexible structure increase due to elastic deformation. However, additional research is needed to suggest general guidelines for optimal design of the flexible structure.
2. The minimum film thickness is the largest when the film thickness distribution in the area that is flexible and where the oil film pressure is generated is almost constant along the axial direction.
3. The minimum film thickness increases effectively when the thickness of the flexible structure is about 0.4 times the shaft radius under a constant load.
4. As the dimensionless length of the flexible structure increases, the minimum film thickness increases up to a certain length and then decreases. The reason for this is that an appropriate length of the flexible structure is advantageous for obtaining the oil film thickness based on the elastic deformation region.
5. The rectangular-shaped flexible structure is more effective for increasing the minimum film thickness than the taper-shaped one for all tilting ratios. The reason for this is that it is not easy to increase the oil film thickness based on the elastic deformation of the inside of the flexible structure.
6. This study also showed that application of the flexible structure can improve the lubrication characteristics of misaligned journal bearing; however, further studies are needed on the analyses of abnormal operating conditions, such as shock or fluctuating load conditions, and optimization of the flexible structure.

**Author Contributions:** Conceptualization, S.-H.H. and W.-J.J.; literature review and formal analysis, W.-J.J.; writing-original draft preparation, methodology, W.-J.J.; review and editing, S.-H.H.; funding acquisition, S.-H.H. All authors have read and agreed to the published version of the manuscript.

**Funding:** This research received no external funding.

**Data Availability Statement:** Not applicable.

**Acknowledgments:** This work was supported by Korea Hydro & Nuclear Power Co. (2022).

**Conflicts of Interest:** The authors declare no conflict of interest.

## Nomenclature

| | |
|---|---|
| $A$ | Dimensionless thickness of the flexible the structure ($=a/r$) |
| $E$ | Young's modulus [GPa] |
| $E^*$ | Dimensionless Young's modulus ($=c^2 E/(6r^2 \eta \omega)$) |
| $F$ | Dimensionless force ($=c^2 f/(6r^4 \eta \omega)$) |
| $F_O$ | Dimensionless oil film force ($=c^2 f_o/(6r^4 \eta \omega)$) |
| $F_X$ | Component of dimensionless force in the $X$ direction |
| $F_Y$ | Component of dimensionless force in the $Y$ direction |
| $F_Z$ | Component of dimensionless force in the $Z$ direction |
| $F_{OX}$ | Component of dimensionless oil film force in the $X$ direction |
| $F_{OY}$ | Component of dimensionless oil film force in the $Y$ direction |
| $H$ | Dimensionless oil film thickness ($=h/c$) |
| $H_e$ | Dimensionless oil film thickness variation by elastic deformation ($=h_e/c$) |
| $H_{min}$ | Dimensionless minimum film thickness ($=h_{min}/c$) |
| $L$ | Ratio of length to radius of the bearing ($=l/r$) |
| $L_f$ | Dimensionless length of the flexible structure ($=l_f/l$) |
| $O$ | Center of the shaft at the middle of the bearing |
| $O_1, O_2$ | Centers of the shaft at both ends of the bearing |
| $P$ | Dimensionless oil film pressure ($=c^2 (p - p_a)/(6r^2 \eta \omega)$) |
| $P_b$ | Dimensionless pressure at the bearing ends and oil feeding groove ($=c^2 (p_b - p_a)/(6r^2 \eta \omega)$) |
| $U$ | Dimensionless displacement in the element ($=u/r$) |
| $W$ | Dimensionless load acting on the shaft ($=c^2 w/(6r^4 \eta \omega)$) |
| $X, Y, Z$ | Dimensionless rectangular coordinate system ($X = x/r$, $Y = y/r$, $Z = z/r$) |
| $a$ | Thickness at the outer end of the flexible structure [mm] |
| $c$ | Clearance [μm] |
| $d$ | Thickness at the inner end of the flexible structure [mm] |
| $e$ | Eccentric amount [μm] |
| $e'$ | Tilting amount of the shaft [μm] |
| $f$ | Force [N] |
| $f_o$ | Oil film force [N] |
| $f_x$ | Component of force in the $x$ direction [N] |
| $f_y$ | Component of force in the $y$ direction [N] |
| $f_z$ | Component of force in the $z$ direction [N] |
| $f_{ox}$ | Component of oil film force in the $x$ direction [N] |
| $f_{oy}$ | Component of oil film force in the $y$ direction [N] |
| $h$ | Oil film thickness [μm] |
| $h_e$ | Change in oil film thickness due to elastic deformation [μm] |
| $h_{min}$ | Minimum film thickness [μm] |
| $l$ | Length of the bearing [mm] |
| $l_f$ | Length of the flexible structure [mm] |
| $p$ | Oil film pressure [MPa] |
| $p_a$ | Atmospheric pressure [MPa] |
| $p_b$ | Pressure at bearing ends and oil feeding groove [MPa] |
| $r$ | Radius of the bearing [mm] |
| $u$ | Displacement in the element [mm] |
| $w$ | Load acting on the shaft [N] |
| $x, y, z$ | Rectangular coordinate system [mm] |
| $\beta$ | Ratio of clearance to bearing radius ($=c/r$) |
| $\varepsilon$ | Eccentricity ratio ($=e/c$) |
| $\varepsilon'$ | Tilting ratio ($=e'/c$) |
| $\gamma$ | Ratio of thickness at both ends of the flexible structure ($=d/a$) |
| $\eta$ | Absolute viscosity of the lubricant [Pa·s] |
| $\nu$ | Poisson's ratio |

| $\theta$ | Cylindrical coordinate [rad] |
|---|---|
| $\theta_W$ | Direction of the load in the cylindrical coordinate system [rad] |
| $\omega$ | Angular velocity [rad/s] |
| $\psi$ | Attitude angle [rad] |

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
