# Peer review of "A New Type of Misaligned Journal Bearing with Flexible Structure"

_lubricants, doi:10.3390/lubricants11060256_

Round 1
Reviewer 1 Report
Comments to the Authors:
In this manuscript, the authors proposed a flexible structure to improve the lubrication performances of the misaligned journal bearings. After a careful review of the manuscript, my opinion is minor revision, and the comments are listed as follows:
1. Most of the references in this paper are before 2015, a survey on the latest researches related to this topic is suggested.
2. Please define all the variables when they appear for the first time in the main text and the first equation, e.g., equations (1), (2) and (3).
3. All variables in the main text need to be in italics and keep the consistent format with which in the equations, e.g., ‘h’ in line 158, ‘pa’ in line 142.
4. There are spelling mistakes in the manuscript, please double check the whole text, e.g., ‘altitude’ in line 163. In addition, the attitude angle ψ does not appear in the section of nomenclature, please double check to ensure that all the parameters in the manuscript are included in Nomenclature.
5. The center of the bearing is suggested to be marked in Figs. 2(a) and 3(a).
Author Response
Dear Editor and Reviewer
We would like to thank the reviewers for their thorough reviews and useful comments and insights. Changes suggested by the reviewers have been incorporated in the manuscript. I hope that the referees will find them satisfactory. If there are further changes to be made, we will be happy to comply. We prepared sincere responses to your comments as follows.
……………………………………………………………………………………………………………………….
In this manuscript, the authors proposed a flexible structure to improve the lubrication performances of the misaligned journal bearings. After a careful review of the manuscript, my opinion is minor revision, and the comments are listed as follows:
Point 1: Most of the references in this paper are before 2015, a survey on the latest researches related to this topic is suggested.
Response 1: We added the latest references related to our research as follows.
- Cai, J.; Xiang, G.; Li, S.; Guo, J.; Wang J.; Chen, S.; Yang, T. Mathematical modeling for non-linear dynamic mixed friction behaviors of novel coupled bearing lubricated with low viscosity fluid. Phys. Fluids, 2022, 34, 093612. https://doi.org/10.1063/5.0108943.
- Xiang, G.; Yang, T.; Guo, J.; Wang, J.; Liu, B.; Chen, S. Optimization transient wear and contact performances of water-lubricated bearings under fluid-solid-thermal coupling condition using profile modification. Wear, 2022, 502-503, 204379. https://doi.org/10.1016/j.wear.2022.204379.
- Zhou, Y., Wang, Y.; Zhao, J. Influence on journal bearing considering wall-slip in EHL. IOP Conf. Ser.: Mater, 2018, 394, 042042. doi:10.1088/1757-899X/394/4/042042.
Point 2: Please define all the variables when they appear for the first time in the main text and the first equation, e.g., equations (1), (2) and (3).
Response 2: We added explanations about variables as follows.
where, A, L, Lf, β and γ express dimensionless thickness of the flexible the structure, ratio of length to radius of the bearing, dimensionless length of the flexible structure, ratio of clearance to bearing radius and ratio of thickness at both ends of the flexible structure.
where, r, z, and θ express cylindrical coordinate. Moreover, h, p, η and ω present oil film thickness, oil-film pressure, absolute viscosity of the lubricant and angular velocity.
where H, P and Z represent dimensionless oil film thickness, dimensionless oil film pressure and dimensionless rectangular coordinate system in the z direction.
Moreover, we added and corrected expression in Nomenclature as follows.
e : Eccentric amount [μm]
θW : Direction of the load in the cylindrical coordinate system [rad]
ψ : Attitude angle [rad]
Point 3: . All variables in the main text need to be in italics and keep the consistent format with which in the equations, e.g., ‘h’ in line 158, ‘pa’ in line 142.
Response 3: We revised in italics including line 142, 158.
Point 4: There are spelling mistakes in the manuscript, please double check the whole text, e.g., ‘altitude’ in line 163. In addition, the attitude angle ψ does not appear in the section of nomenclature, please double check to ensure that all the parameters in the manuscript are included in Nomenclature.
Response 4: We revised the misspelling and we corrected in Nomenclature.
altitude -> attitude
e : Eccentric amount [μm]
θW : Direction of the load in the cylindrical coordinate system [rad]
ψ : Attitude angle [rad]
Point 5: The center of the bearing is suggested to be marked in Figs. 2(a) and 3(a).
Response 5: We expressed the center of bearing in Figs. 2(a) and 3(a).
We thank the reviewer again for their time and insights.
Sincerely yours,
Corresponding author Sung-Ho Hong

Reviewer 2 Report
This paper describes the effect of the specifications of the flexible structure on minimum film thickness of the journal bearings under misaligned shaft. Reviewer think the reserch results are interesting and valuable.
Minor revision points are shown below.
P.5 L.2 The mesh sizes are mentioned. However, there is no writing they are correct or not. Did you check the correctness of the sizes? It would be better to mention about it.
P.6 L.4 the oil feeding groove is mentioned, however I couldn't find it in the figures. It would be better to show it in the figure 2 or 3.
The deformation of the flexible structure is important in the results of this paper. The thickness variation is described in detail, but it would be good to show how the deformation, especially in the axial direction, changes with changes in A and Lf. It would be interesting to see how the deformation changes at around 0.4. Please consider.
Author Response
Dear Editor and Reviewer
We would like to thank the reviewers for their thorough reviews and useful comments and insights. Changes suggested by the reviewers have been incorporated in the manuscript. I hope that the referees will find them satisfactory. If there are further changes to be made, we will be happy to comply. we prepared sincere responses to your comments as follows.
……………………………………………………………………………………………………………………….
This paper describes the effect of the specifications of the flexible structure on minimum film thickness of the journal bearings under misaligned shaft. Reviewer think the research results are interesting and valuable.
Minor revision points are shown below.
Point 1: P.5 L.2 The mesh sizes are mentioned. However, there is no writing they are correct or not. Did you check the correctness of the sizes? It would be better to mention about it.
Response 1: We added an explanation as follows.
A mesh validation was performed that the difference in the results values was less than 2%.
Point 2: P.6 L.4 the oil feeding groove is mentioned, however I couldn't find it in the figures. It would be better to show it in the figure 2 or 3.
Response 2: We added a groove of oil feeding in Figs 2(a) and 3(a). Moreover, we added center of the bearing.
Point 3: The deformation of the flexible structure is important in the results of this paper. The thickness variation is described in detail, but it would be good to show how the deformation, especially in the axial direction, changes with changes in A and Lf. It would be interesting to see how the deformation changes at around 0.4. Please consider.
Response 3: We appreciate your comment. We think that distributions of deformation with variation of A and Lf were expressed in Fig. 13. There may be a little lacking, but I think you can check it with this picture. As mentioned in the cover letter, a paper on the impact load of a journal bearing similar to that used in this analysis is being prepared. At this time, we would be considered to prepare figures for a more detailed explanation by reflecting the opinions of the reviewers.
We thank the reviewer again for their time and insights.
Sincerely yours,
Corresponding author Sung-Ho Hong
